# GenRec: Unifying Video Generation and Recognition with Diffusion Models

**Zejia Weng**[1,2], **Xitong Yang**[3], **Zhen Xing**[1,2], **Zuxuan Wu**[1,2†] , **Yu-Gang Jiang**[1,2]

[1] Shanghai Key Lab of Intell. Info. Processing, School of CS, Fudan University
[2] Shanghai Collaborative Innovation Center of Intelligent Visual Computing
[3] Department of Computer Science, University of Maryland

## Abstract

Video diffusion models are able to generate high-quality videos by learning strong spatial-temporal priors on large-scale datasets. In this paper, we aim to investigate whether such priors derived from a generative process are suitable for video recognition, and eventually joint optimization of generation and recognition. Building upon Stable Video Diffusion, we introduce GenRec, the first unified framework trained with a random-frame conditioning process so as to learn generalized spatial-temporal representations. The resulting framework can naturally supports generation and recognition, and more importantly is robust even when visual inputs contain limited information. Extensive experiments demonstrate the efficacy of GenRec for both recognition and generation. In particular, GenRec achieves competitive recognition performance, offering 75.8% and 87.2% accuracy on SSV2 and K400, respectively. GenRec also performs the best on class-conditioned image-to-video generation, achieving 46.5 and 49.3 FVD scores on SSV2 and EK-100 datasets. Furthermore, GenRec demonstrates extraordinary robustness in scenarios that only limited frames can be observed. Code will be available at https://github.com/wengzejia1/GenRec.

## 1 Introduction

Diffusion models have achieved significant success in the field of image and video generation over the past few years. A variety of generative tasks have been revolutionized by using diffusion models trained on Internet-scale data, such as text-to-image generation [33, 30], image editing [23], and more recently, text-to-video generation [15, 2, 52] and text&image-to-video generation [56, 18, 21]. The excellent generative capabilities of diffusion models suggest that informative representation is learned during the generative training and strong visual priors are captured by the backbone models [9, 43, 6]. Therefore, recent work has explored leveraging the image diffusion models for image understanding tasks, including image recognition [9, 8], object detection [7, 57], segmentation [55] and correspondence mining [39]. However, the capability of *video diffusion* models to effectively capture spatial-temporal information is not fully understood, and their potential for downstream video understanding tasks remains under-explored.

In this paper, we study the potential of video diffusion models [28, 1, 27], particularly the unconditioned or image-conditioned models, for video understanding by addressing the three key problems: (a) Does the backbone model trained for video generation extract effective spatial-temporal representations for semantic video recognition? (b) Can we retain the video generation capability by jointly optimizing generation and recognition? (c) Will such a unified training framework further benefit video understanding, especially in noisy scenarios where only limited frames are available [3, 25].

---

[†]Corresponding author.

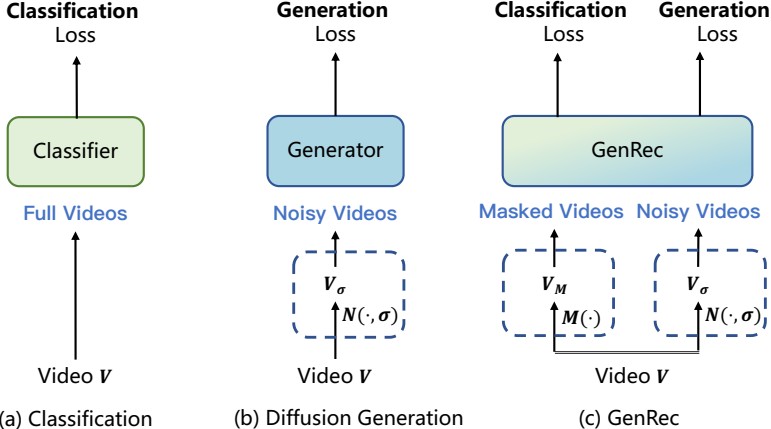

Figure 1: Comparison of classical pipelines for video classification and generation tasks with our proposed GenRec method. **(a) Classification**: Typical video classification focus on understanding complete videos. **(b) Diffusion Generation**: Diffusion models learn the noise reduction trajectory from videos with varying levels of noise. These two distinct training paradigms present challenges for task unification. To bridge this gap, we propose **(c) GenRec**: a learning framework that processes mask frames $V_M$ using a masking function $M(\cdot)$ and noise videos $V_\sigma$ with noise sampling $\mathcal{N}(\cdot, \sigma)$, aiming to simultaneously learn video understanding and content completion with the same partially observed visual content.

While conceptually appealing, unifying video generation and recognition into a diffusion framework is non-trivial. Prior work either views the diffusion models as frozen feature extractors [9, 55, 39], or deconstructs them for new tasks while sacrificing their original generation capability [8]. One major challenge comes from their distinct training and inference processes. Diffusion models are typically optimized using corrupted inputs, optionally augmented with a single conditioning frame, to achieve unconditioned or image-conditioned generation during inference [27, 1]. In contrast, video recognition models require access to multiple frames to reason about temporal relationships and expect clean inputs during inference [49, 51, 48]. Consequently, training a recognition model using corrupted videos and single-image conditions tends to suffer from inferior model optimization and a more significant training-inference gap.

To this end, we propose GenRec, a unified video diffusion model that enables joint optimization for video generation and recognition. Our model is built upon the open-source, image-conditioned Stable Video Diffusion model (SVD) [1], which encodes strong spatial-temporal priors by pretraining on large-scale image and video data. However, instead of conditioning on the same image across all video frames, we propose to condition on a random subset of frames while masking the remaining ones (see Figure 2). This simple random-frame conditioning process effectively bridges the gap between the learning processes of the two tasks. On the one hand, the generation capability of SVD is extended to handle arbitrary frame prediction, which provides more flexible and unambiguous video generation. On the other hand, conditioning on a random subset of frames allows the model to learn more discriminative and robust features for the recognition task. As shown in Figure 1, the model is jointly optimized using both generative supervision (*i.e.*, noise prediction) and classification supervision.

We conduct extensive experiments to evaluate the performance of GenRec for both recognition and generation. Without sacrificing the generation capabilities, GenRec demonstrates competitive video recognition performance, offering 75.8% and 87.2% accuracy on SSV2 and K400, respectively. Furthermore, GenRec demonstrates extraordinary robustness in scenarios that only limited frames can be observed. For example, when only the front half of the video can be observed, GenRec achieves the 57.7% accuracy, which corresponds to 76.6% of the accuracy (75.3%) when the entire video is visible, emonstrating a higher accuracy retention ratio than other methods. By leveraging the recognition model for classifier guidance [11], GenRec also achieves superior class-conditioned image-to-video generation results, with FVD scores of 46.5 and 49.3 on the SSV2 and EK-100 datasets, respectively.

## 2 Preliminary

Representing the data distribution as $p_{\text{data}}(\mathbf{z})$ with a standard deviation of $\sigma_{\text{data}}$, we can obtain a family of smoothed distributions $p(\mathbf{z}; \sigma)$ by adding independent and identically distributed Gaussian noise with standard deviation $\sigma$. In the spirit of diffusion models, the generation process begins with a noise image $\mathbf{z}_N \sim N(0, \sigma_{\text{max}}^2 \mathbf{I})$ and iteratively denoises it at decreasing noise levels $\sigma_N = \sigma_{\text{max}} > \sigma_{N-1} > \ldots > \sigma_0 = 0$. The final denoised result $\mathbf{z}_0$ is thus distributed according to the original data.

In the EDM [22] framework, the original $\mathbf{z}_0$ will be diffused as:

$$\mathbf{z}_\sigma = \mathbf{z}_0 + \sigma \cdot N(0, \mathbf{I}), \tag{1}$$

and the corresponding PF-ODE [34] follows:

$$d\mathbf{z}_\sigma = -\sigma \cdot \nabla_{\mathbf{z}} \log p_\sigma(\mathbf{z}_\sigma) d\sigma, \tag{2}$$

where $\nabla_{\mathbf{z}} \log p_t(\mathbf{z}_t)$ is the score function. Noise schedule $\sigma(t)$ is set as time step $t$. The training objective is to minimize the $L2$ loss with the denoiser network $D_\theta$ for different $\sigma$:

$$\mathbb{E}_{\mathbf{z}_0 \sim p_{data}} ||D_\theta(\mathbf{z}_\sigma) - \mathbf{z}_0||_2^2, \tag{3}$$

with the relation between $D_\theta$ and the score function $\nabla_{\mathbf{z}} \log p(\mathbf{z}; \sigma)$ as follows:

$$\nabla_{\mathbf{z}} \log p(\mathbf{z}; \sigma) = (D(\mathbf{z}_\sigma) - \mathbf{z})/\sigma^2. \tag{4}$$

SVD [31] utilizes the EDM framework to perform generative training on large-scale video datasets, resulting in a high-quality video generation model. An image-to-video generation model capable of forecasting future frames given the first frame has been released. Following SVD method, we also process videos in latent space. Given an input video $\mathbf{V} \in \mathbb{R}^{T \times H \times W \times 3}$, a pretrained VAE encoder is used to project it into the latent space frame by frame, resulting in the latent representation $\mathbf{z}_0 \in \mathbb{R}^{T \times h \times w \times D}$. We then build GenRec based on SVD, inheriting its strong spatial-temporal priors as foundation for the subsequent generation and classification tasks.

## 3 GenRec

We now introduce GenRec, a simple yet efficient framework, that can not only generate temporally-coherent videos conditioned on an arbitrary number of provided frames but also is able to recognize actions and events with the help of encoded spatial-temporal priors. To this end, GenRec explores the strong spatial-temporal priors learned by a video diffusion model. In this work, we instantiate GenRec with the powerful open-source Stable Video Diffusion model (SVD) [1], which is pretrained on large-scale video datasets and is able to produce a photo-realistic video when provided a single frame. Then, for generation, GenRec follows the classical EDM framework to learn noise reduction trajectories. For recognition, on the other hand, GenRec operates on intermediate decoded features using a recognition head. Furthermore, to generate videos in a more free fashion, *i.e.* an arbitrary collection of frames used as condition, we design a latent masking strategy that "interpolates" masked frames. Such a strategy also benefits recognition by easing the training process. More importantly, by doing so GenRec supports a multitude of downstream tasks, particularly when limited visual information is provided.

### 3.1 Pipeline Overview

**Latent diffusion and latent masking.** During the diffusion process, the Gaussian noise with a certain noise level is added to the latent representation $\mathbf{z}_0$, creating a noisy latent representation $\tilde{\mathbf{z}}_i$ following Equation (1). Recall that while SVD contains powerful spatial-temporal priors, it can only perform generation when the first frame is provided. To allow a more "free" generation with an arbitrary number of frames as inputs, we design a latent masking strategy. More specifically, we apply a random mask $\mathbf{m}$ to the latent representation $\mathbf{z}_0$, producing a masked latent representation $\overline{\mathbf{z}_0}$. Such a strategy encourages the model to reconstruct the original video content from incomplete frames, which is in a similar spirit to MAE [19]. Note that when only the first latent is available, it degrades to the same as SVD; if all latents are masked out, this degrades to unconditional generation. Furthermore, doing so also benefits recognition tasks when limited visual clues are available. For example, in scenarios with limited bandwidth leading to reduced frame rates, the ability of video

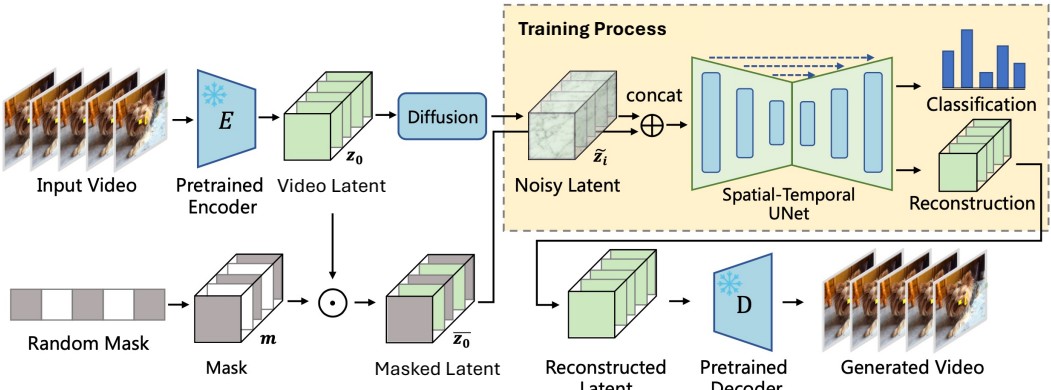

Figure 2: The pipeline of our proposed video processing method. The input video is first processed by a pretrained encoder $E$ to produce a latent representation $\mathbf{z}_0$, then undergoes diffusion to generate a noisy latent $\tilde{\mathbf{z}}_t$. The random mask $\mathbf{m}$ is used to create the masked latent $\overline{\mathbf{z}_0}$. During training, the noisy latent is concatenated with the masked latent as condition and fed into a Spatial-Temporal UNet, resulting in both reconstruction and recognition outputs. The reconstructed latent can be decoded by the pretrained decoder $D$ to produce the final generated video.

frame complementation enables the model to better predict and perceive complete video information. In practice, we simulate such conditions by randomly erasing half to all video conditions, retaining on average only about one-fourth of the original video information. This technique allows the model to effectively fill in the missing information, enhancing its ability to recognize and understand the video content despite the reduced data availability.

**Unifying generation and understanding.** To unify generation with masked latents, GenRec predicts pixel-level and semantic-level contents with the combination of the noisy latent $\tilde{\mathbf{z}}_i$ and the masked latent $\overline{\mathbf{z}_0}$ gained by the aforementioned latent diffusion and latent masking. The two latents are channel-wise concatenated $[\tilde{\mathbf{z}}_i, \overline{\mathbf{z}_0}]$ and are fed into a Spatial-Temporal UNet, together with features from observed frames, to learn spatial and temporal representations, following [1]. The weights of the UNet are initialized from [1] to obtain spatial and temporal priors, learned on large-scale video datasets.

For the generation task, the UNet aims to reconstruct the original latent representation from the combined noisy and masked inputs. Representing UNet as the mapping function $F_\theta$, its goal is to predict clean latent, which, according to the EDM framework, takes the form of a representation mapping as follows:

$$D_\theta(\tilde{\mathbf{z}}_i; \overline{\mathbf{z}_0}, \sigma) = c_{skip}(\sigma)\tilde{\mathbf{z}}_i + c_{out}(\sigma)F_\theta([c_{in}(\sigma)\tilde{\mathbf{z}}_i, \overline{\mathbf{z}_0}]), \tag{5}$$

in which we set the same skip connection $c_{skip}$, scaling factor $c_{out}$ and $c_{in}$ as [1].

For the recognition task, we break down the UNet mapping function as $F = F_{tail} \cdot F_{head}$. And we consider $F_{head}([c_{in}(\sigma)\tilde{\mathbf{z}}, \overline{\mathbf{z}}]) \in \mathbb{R}^{T \times h' \times w' \times D'}$ as the compact video representation extracted from the intermediate layer of the UNet model, which is then fed into the classifier head $\phi_\theta$, consisting of an attentive pooler and a fully connected layer to predict video categories:

$$\hat{\mathbf{y}} = \phi(F_{head}([c_{in}(\sigma)\tilde{\mathbf{z}}_i, \overline{\mathbf{z}_0}])). \tag{6}$$

## 3.2 Optimization

We train GenRec with both generation and classification objectives, encouraging the model to learn high-quality video generation and accurate video understanding.

The generative loss uses a $L2$ loss to measure the difference between the original latent representation and the reconstructed output produced by the UNet, and is defined as:

$$L_G(\mathbf{z}_0, \tilde{\mathbf{z}}_i, \overline{\mathbf{z}_0}; \sigma) = \lambda(\sigma)\|D_\theta(\tilde{\mathbf{z}}_i; \overline{\mathbf{z}_0}, \sigma) - \mathbf{z}_0\|^2, \tag{7}$$

where $D_\theta(\tilde{\mathbf{z}}_i; \overline{\mathbf{z}_0}, \sigma)$ is the denoised output mentioned in Equation (5), and $\lambda(\sigma)$ is a weighting function based on the noise level $\sigma$ referring to [1, 22]. While the classification loss uses a cross-entropy loss to measure the discrepancy between the true labels and the predicted labels, and is defined as:

$$L_D(\mathbf{y}, \hat{\mathbf{y}}) = -\sum_i \mathbf{y}_i \log(\hat{\mathbf{y}}_i), \tag{8}$$

where $\mathbf{y}$ denotes the ground truth labels, and $\hat{\mathbf{y}}$ represents the predicted labels referring to Equation (6).

To balance the learning of generative and recognition tasks, we set a balancing weight $\gamma$ to control the relative importance of each loss in the overall objective function. The total loss $L$ is given by:

$$L = L_D + \gamma L_G, \tag{9}$$

## 3.3 Inference for Different Downstream Tasks

With the above training strategies, we now introduce how GenRec can flexibly support different types of generation and recognition tasks.

**Video generation conditioned on frames.** Once trained, GenRec is able to generate high-quality videos conditioned on an arbitrary number of given frames, thanks to the latent masking strategy. Particularly, following the EDM stochastic sampler framework and Equation (2), GenRec iteratively denoises the video conditioned on the masked latent $\overline{\mathbf{z}_0}$, as shown below:

$$\mathbf{z}_{i-1} = \hat{\mathbf{z}}_i + \epsilon_\theta(\hat{\mathbf{z}}_i; \overline{\mathbf{z}_0}) = \hat{\mathbf{z}}_i + (t_{i-1} - \hat{t}_i)\frac{d\hat{\mathbf{z}}_i}{d\hat{t}_i} \tag{10}$$

$$= \hat{\mathbf{z}}_i + (t_{i-1} - \hat{t}_i)(-1)\frac{D_\theta(\hat{\mathbf{z}}_i; \overline{\mathbf{z}_0}, \sigma) - \hat{\mathbf{z}}_i}{\hat{t}_i}, \tag{11}$$

where $\hat{\mathbf{z}}_i$ is derived from $\tilde{\mathbf{z}}_i$ adding a perturbation. With an iteratively denoising process, we can finally obtain the denoised video latent $\mathbf{z}_0$ which can be decoded as a complete video.

**Video generation conditioned on classes.** When the number of visible frames is extremely limited, the motion trajectory becomes unpredictable and thus it would be hard to make a reliable prediction of the future. To mitigate this issue, GenRec supports adding category information to guide video generation in the expected desired direction.

Formally, we simplify Equation (11) with Equation (4), and obtain:

$$\epsilon_\theta(\hat{\mathbf{z}}_i; \overline{\mathbf{z}_0}) = (t_{i-1} - \hat{t}_i)(-1)\frac{D_\theta(\hat{\mathbf{z}}_i; \overline{\mathbf{z}_0}, \sigma) - \hat{\mathbf{z}}_i}{\hat{t}_i} \tag{12}$$

$$= (-1)(t_{i-1} - \hat{t}_i)\hat{t}_i \nabla_{\hat{\mathbf{z}}_i} \log p_\theta(\hat{\mathbf{z}}_i). \tag{13}$$

We substitute the score function $\nabla_{\hat{\mathbf{z}}_i} \log p_\theta(\hat{\mathbf{z}}_i)$ with the conditional form $\nabla_{\hat{\mathbf{z}}_i} \log p_\theta(\hat{\mathbf{z}}_i|y)$, in which $y$ denotes the conditional class. By applying Bayes' Theorem, the original score function can be replaced by $p(\hat{\mathbf{z}}_i)p(y|\hat{\mathbf{z}}_i)$, and we can get the conditional version of residual, denoted as $\epsilon_\theta^*(\hat{\mathbf{z}}_i; \overline{\mathbf{z}_0})$:

$$\epsilon_\theta^*(\hat{\mathbf{z}}_i; \overline{\mathbf{z}_0}) = (-1)(t_{i-1} - \hat{t}_i)\hat{t}_i \nabla_{\hat{\mathbf{z}}_i} \log p(\hat{\mathbf{z}}_i)p(y|\hat{\mathbf{z}}_i) \tag{14}$$

$$= (-1)(t_{i-1} - \hat{t}_i)\hat{t}_i[\nabla_{\hat{\mathbf{z}}_i} \log p(\hat{\mathbf{z}}_i) + \nabla_{\hat{\mathbf{z}}_i} \log p(y|\hat{\mathbf{z}}_i)] \tag{15}$$

$$= \epsilon_\theta(\hat{\mathbf{z}}_i; \overline{\mathbf{z}_0}) - (t_{i-1} - \hat{t}_i)\hat{t}_i \nabla_{\hat{\mathbf{z}}_i} \log p(y|\hat{\mathbf{z}}_i) \tag{16}$$

Considering the scaling factor of $\hat{\mathbf{z}}_i$: $c_{in}(\sigma) = \frac{1}{\sqrt{\sigma^2 + \sigma_{data}}}$ (following [22], and $\sigma_i = t_i$), that would pre-scale the input as $c(\mathbf{z}_i) = c_{in}(t_i) \cdot \mathbf{z}_i$ before model processing, the formulation can be further transferred as:

$$\epsilon_\theta^*(\hat{\mathbf{z}}_i; \overline{\mathbf{z}_0}) = \epsilon_\theta(\hat{\mathbf{z}}_i; \overline{\mathbf{z}_0}) - \frac{(t_{i-1} - \hat{t}_i)\hat{t}_i}{\sqrt{\hat{t}_i^2 + \sigma_{data}}} \nabla_{c(\hat{\mathbf{z}}_i)} \log p(y|c(\hat{\mathbf{z}}_i)) \tag{17}$$

Following [11], we sharpen the distribution of $p(y|\mathbf{z})$ by multiplying a scaling factor $s > 1$, shown as $s \cdot \nabla_{\mathbf{z}} \log p(y|\mathbf{z}) = \nabla_{\mathbf{z}} \log \frac{1}{Z}p(y|\mathbf{z})^s$ where $Z$ is an arbitrary constant. Larger scaling value would

bring more attention to the target category. Here, $p(y|c(\hat{\mathbf{z}}_i))$ comes from the classification branch in GenRec. Finally, we can use the same EDM sampling procedure with the derived class information to generate samples.

**Standard video recognition.** Based on Equation (6), GenRec can do the classical video recognition by setting constant no-mask, and thus $\overline{\mathbf{z}_0}$ is replaced by $\mathbf{z}_0$ and the prediction follows:

$$\hat{\mathbf{y}} = \phi(F_{head}([c_{in}(\sigma)\tilde{\mathbf{z}}_i, \mathbf{z}_0])). \tag{18}$$

**Video recognition with partially observed frames.** Based on Equation (6), GenRec can be applied to video recognition with partially observed frames, *e.g.*, early action prediction that aims to predict future events based on the initial frames, sparse video recognition where videos are sparsely encoded and transmitted due to bandwidth limitations. By masking the invisible frames to get $\tilde{\mathbf{z}}_i$, and replacing the noisy latent with random noise $\sim$ obeying Gaussian distribution, GenRec can do the prediction for partially visible videos, following:

$$\hat{\mathbf{y}} = \phi(F_{head}([\sim, \overline{\mathbf{z}_0}])). \tag{19}$$

## 4 Experiments

### 4.1 Experimental Setup

**Datasets.** In our experiments, we use the following four datasets: Something-Something V2 (SSV2) [17], Kinetics-400 (K400) [24], UCF-101 [35] and Epic-Kitchen-100 (EK-100) [10]. SSV2 dataset is designed for fine-grained action recognition and it contains 174 action classes, 220,847 short video clips with an average duration of 4 seconds. K400 contains 400 action classes, 306,245 video clips with an average duration of 10 seconds. The UCF-101 dataset comprises 13,320 videos from 101 action categories and is widely utilized for human action recognition. The EK-100 dataset focuses on egocentric vision. It contains a total of 90,000 annotated action segments, encompassing 97 verb classes and 300 noun classes.

**Evaluation protocols.** GenRec performs both generation and recognition tasks. For generation, we use the Fréchet Video Distance (FVD) [41] metric to assess the quality of the generated videos. A lower FVD score indicates higher fidelity and realism. For recognition, we measure the top-1 accuracy that reflects the portion of correctly classified videos. We validate our model performance in formal video recognition, partial video recognition, class-conditioned image-to-video generation and frame completion with the above metrics.

**Implementation details.** We initially set the learning rate to $1.0 \times 10^{-5}$ and set the total batch size as 32. Only generation loss will be retained for model adaptation on specific datasets. We train 200k steps on EK-100 and UCF, and 300k steps on SSV2 and K400, respectively. Subsequently, we finetune GenRec with both generation and recognition losses. The learning rate is set to $1.25 \times 10^{-5}$ and decayed to $2.5 \times 10^{-7}$ using a cosine decay scheduler. We warm up models with 5 epochs, during which the learning rate is initially set as $2.5 \times 10^{-7}$ and linearly increases to the initial learning rate $1.25 \times 10^{-5}$. The loss balance ratio $\gamma$ is set to 10, and the learning rate for the classifier head is ten times higher than the base learning rate. We drop out the conditions 10% of the time for supporting classifier-free guidance [20], and we finetune on K400 for 40 epochs and 30 epochs on other datasets. The training is executed on 8 A100s and each contains a batch of 8 samples. We sample 16 frames for each video.

### 4.2 Main Results

**Comparison to state-of-the-art in video recognition and video generation.** We compare with state-of-the-art methods in terms of their recognition accuracy and generation quality. The results are summarized in Table 1. The first two blocks of the table presents current advanced video recognition models, while the third block demonstrates the performance of the diffusion-based class-guided image-to-video generation.

As shown in the table, GenRec achieves optimal results or performs on par with the state-of-the-art approaches. In terms of video recognition, GenRec achieves 75.8% accuracy on SSV2 dataset,

Table 1: Performance of Video Recognition and Generation Methods. We evaluate on video recognition and class-conditioned image-to-video generation tasks. SEER† predicts 16 frames, while others predict 12 frames. Top-1 accuracy and FVD scores are reported. Baseline I adapts SVD to datasets with generative fine-tuning and then uses attentive-probing for classification. Baseline II fully finetunes SVD with classification supervision only in traditional classification framework.

| Method | Resolution | Param. | Classification Acc (↑) | | Generation FVD (↓) | |
| --- | --- | --- | --- | --- | --- | --- |
| | | | SSV2 | K400 | SSV2 | EK-100 |
| *w/o multi-modal align.* | | | | | | |
| VideoMAE-L [40] | 224×224 | 305M | 74.3 | 85.2 | - | - |
| VideoMAE-H [40] | 224×224 | 633M | - | 86.6 | - | - |
| OmniMAE-H [16] | 224×224 | 650M | 75.5 | 85.4 | - | - |
| MVD-H [44] | 224×224 | 633M | 77.3 | 87.2 | - | - |
| Hiera-L [32] | 224×224 | 214M | 75.1 | 87.3 | - | - |
| Hiera-H [32] | 224×224 | 673M | - | 87.8 | - | - |
| MaskFeat-L [47] | 312×312 | 218M | 75.0 | 86.4 | - | - |
| *w/ multi-modal align.* | | | | | | |
| InternVideo [45] | 224×224 | 1.3B | 77.2 | 91.1 | - | - |
| InternVideo2 [46] | 224×224 | 6B | 77.4 | 92.1 | - | - |
| OmniVec [36] | - | - | 85.4 | 91.1 | - | - |
| OmniVec-2 [37] | - | - | 86.1 | 93.6 | - | - |
| TATS [14] | 128×128 | - | - | - | 428.1 | 920.0 |
| MCVD [42] | 256×256 | ₊3.5B | - | - | 1407 | 4804 |
| SimVP [13] | 64×64 | - | - | - | 537.2 | 1991 |
| VideoFusion [28] | 256×256 | 1.8B | - | - | 163.2 | 349.9 |
| Tune-A-Video [50] | 256×256 | ₊860M | | | 291.4 | 365.0 |
| SEER [18] | 256×256 | ₊860M | - | - | 112.9 | 271.4 |
| SEER† [18] | 256×256 | ₊860M | - | - | 355.4 | - |
| Baseline I | 256×256 | 2.1B | 63.7 | 82.0 | 50.3 | 53.6 |
| Basline II | 256×256 | 1.9B | 75.9 | 86.6 | - | - |
| GenRec | 256×256 | 2.1B | 75.8 | 87.2 | 46.5 | 49.3 |

surpassing the majority of current state-of-the-art methods. On K400, GenRec achieves 87.2% accuracy, which is on par with the performance of MVD-H (87.2%) and Hiera (87.3%, 87.8%), and surpasses other advanced methods. These results indicate the effectiveness of our approach in video recognition. In addition, GenRec shows a slight performance gap compared to the methods in the second block. It is important to note that these advanced methods benefit significantly from pretraining on large-scale multimodal alignment datasets, which provide extensive cross-modal supervision that enhances their ability to capture semantic relationships across video frames.

We further construct two strong baselines. Baseline I adapts SVD to the respective dataset through generative fine-tuning, followed by attentive-probing for classification, where the backbone is frozen and all frames are used as input. Baseline II involves fully fine-tuning the original SVD model with classification supervision only, ensuring that all frames are visible during training. Compared with them, GenRec performs on par or better. GenRec performs good in supporting not only classification but also generation, demonstrating its comprehensive capability in handling both tasks effectively.

In terms of video generation, we evaluate the model on class-conditioned image-to-video generation task following [18]. Comparing the FVD scores of SEER:112.9 and SEER†:355.4, it can be inferred that generating longer videos with 16 frames is more difficult than generating 12 frames. GenRec generates videos with 16 frames and achieves much lower FVD scores than the other methods, demonstrating the effectiveness of our approach in video generation.

It is worth highlighting that, current research always treats video recognition and generation tasks in a separate manner, and most of the advanced methods focus primarily on either recognition or generation tasks. For instance, SEER method excels in class-conditioned image-to-video generation, but lacks the ability to do video recognition. While current research on representation learning, shown as the first and second blocks in Table 1, lacks the ability to do video generation tasks. In contrast, GenRec not only unifies these tasks, but also achieves competitive results compared to the specialized methods.

Table 2: Early action prediction and limited interpolation problem on Something-Something V2 dataset, with one temporal crop. $\rho$ denotes the visible ratio according to the whole video. Acc denotes to the top-1 accuracy. Ratio metric represent the percentage of maximum performance that the model can maintain at various frame rates. The 'w/o G' experiment refers to the results obtained by removing the generative supervision from our method.

| Method | Metric | Early Action Prediction ($\rho$) | | | | | Limited Inter. Frames | | | |
|---|---|---|---|---|---|---|---|---|---|---|
| | | 0.1 | 0.3 | 0.5 | 0.7 | 1.0 | 2 fs | 3 fs | 4 fs | 16 fs |
| TemPr [38] | Accuracy | 20.5 | 28.6 | 41.2 | 47.1 | 66.3 | - | - | - | - |
| | Retention | 30.9% | 43.1% | 62.1% | 71.3% | 100% | - | - | - | - |
| MVD [44] | Accuracy | - | - | - | - | - | 34.2 | 54.3 | 64.4 | 75.0 |
| | Retention | - | - | - | - | - | 45.6% | 72.4% | 85.9% | 100% |
| MVD† [44] | Accuracy | 26.9 | 39.8 | 55.6 | 70.2 | 75.0 | 53.6 | 65.0 | 68.8 | 75.0 |
| | Retention | 35.9% | 53.1% | 74.1% | 93.2% | 100% | 71.5% | 86.7% | 91.7% | 100% |
| w/o G | Accuracy | 27.3 ↓1.6 | 39.6 ↓2.3 | 56.3↓1.4 | 71.6↓0.8 | 75.0↓0.3 | 53.5↓2.2 | 65.8↓1.1 | 69.8↓1.0 | 75.0↓0.3 |
| GenRec | Accuracy | **28.9** | **41.9** | **57.7** | **72.4** | 75.3 | **55.7** | **67.3** | **70.8** | 75.3 |
| | Retention | **38.4%** | **55.6%** | **76.6%** | **96.1%** | 100% | **74.0%** | **89.4%** | **94.0%** | 100% |

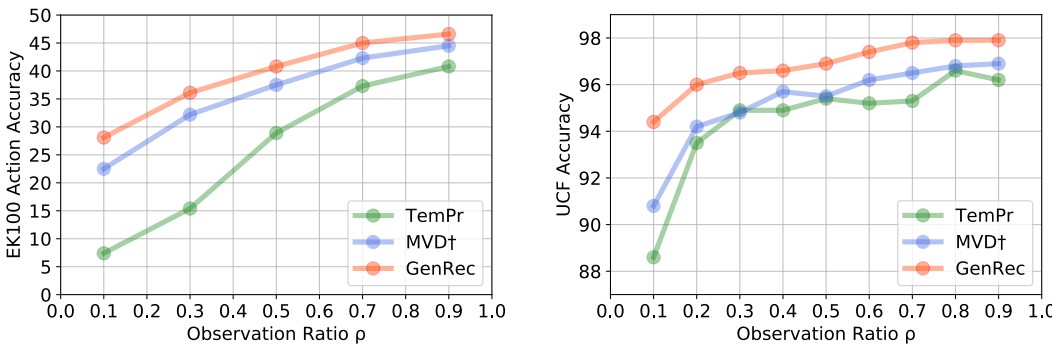

Figure 3: Early action prediction on EK-100 and UCF-100 datasets, with one temporal crop.

**Comparison to state-of-the-art in video recognition with limited frames.** GenRec supports video recognition when only partial frames can be observed. We evaluate this capability on the SSV2 and EK-100 datasets. Our evaluation includes two tasks: an early prediction task, where the model has access only to previous continuous frames following the setting of [38], and a recognition task where videos are sparsely sampled, and the model is expected to make correct predictions. For fair comparisons, we construct two strong baselines. We first apply MVD [44] to directly deal with the recognition task by constructing a dense video through nearest neighbor interpolation. We also construct another baseline similar to our training pipeline, where we apply frame dropout in the training process of MVD [44] for better fitting on task with partial frames, and is named as MVD†. In all settings, the number of fully observed frames is 16.

Table 2 shows the results under these settings, in which $\rho$ denotes the visible ratio (there are a total of 16 frames). In the early action prediction task, GenRec achieves the highest accuracy and ratio metrics at all observation levels. Notably, GenRec and MVD† exhibit similar performance when all frames are observed, but as the number of observed frames decreases, GenRec demonstrates higher accuracy. GenRec also shows superior performance when videos are sparsely sampled, maintaining high accuracy even with fewer observed frames (e.g., 55.7% for 2 frames and 70.8% for 4 frames), indicating its robustness in handling sparse data. Moreover, we compute the ratio metric representing the percentage of maximum performance that the model can maintain at various frame rates, mitigating the unfairness caused by different backbone networks. In this scenario, GenRec still achieves the best performance.

We further investigate the contributions of generation supervision for recognition. As seen in Table 2, removing generation supervision results in noticeable performance degradation across various tasks, especially when the number of visible frames get less. For example, in the early prediction task, the accuracy decreases by 0.3% at $\rho = 1.0$ and by 2.3% at $\rho = 0.3$. These results suggest that generation supervision is essential for maintaining high performance, particularly when the model has to make

Table 3: Relation between generation and recognition.

| Method | Metric | Early Frames | | | | Limited Inter. Frames | | |
|---|---|---|---|---|---|---|---|---|
| | | 2 fs | 5 fs | 8 fs | 12fs | 2 fs | 3 fs | 4 fs |
| GenRec | Acc ↑ | 28.9 | 41.9 | 57.7 | 72.4 | 55.7 | 67.3 | 70.8 |
| | FVD↓ | 57.8 | 44.0 | 30.3 | 16.6 | 46.7 | 31.7 | 24.3 |

Table 4: Choice of UNet layers for feature extraction.

| Up Index | 1 | 2 | 3 |
|---|---|---|---|
| SSV2 | 71.8 | 75.8 | 75.2 |

Table 5: Ablation study on different masking strategies.

| Method | Expectation of masking ratio | SSV2 | |
|---|---|---|---|
| | | Acc (↑) | FVD (↓) |
| GenRec | 75%(our choice) | 75.8 | 46.5 |
| | 50% | +0.3 | +0.5 |
| | 87.5% | -0.9 | -0.3 |

predictions with limited visual information. By incorporating generation supervision, the model can better handle scenarios with incomplete data, improving robustness and accuracy.

We also evaluate the early action prediction on EK-100 and UCF-101. EK-100 is a temporally sensitive dataset similar to SSV2, demanding in terms of the model's temporal modeling capability, while UCF-101 demands more on appearance modeling. We conduct early prediction evaluation on them to further reveal the robustness of our GenRec. As shown in Figure 3, GenRec clearly outperforms TemPr and MVD†. In particular, the improvement becomes more significant as the number of observed frames decreases. More evaluation results can be seen in Appendix A.1.

These results collectively demonstrate that GenRec effectively handles missing video frames. The robustness and high accuracy of GenRec across different datasets and observation ratios highlight its potential for real-world applications where video data might be incomplete or sparsely sampled.

**The relationships between generation and recognition .** We further investigate the consistency between video generation and recognition, as shown in Table 3. We evaluate the performance of video recognition and generation with limited frames and find that the recognition accuracy not only depends on the number of visible frames but also significantly on the location of these frames. Interestingly, uniform sampling appears to facilitate video recognition better than dense sampling from the video prefix. Specifically, with the same number of frames, early prediction consistently shows lower accuracy compared to uniformly sampled frames (e.g., 28.9% vs. 55.7% with 2 frames) and worse FVD scores (e.g., 57.8 vs. 46.7 with 2 frames). When only three interpolated frames are visible, the 31.7 FVD score is comparable to that of an eight-frame prefix (30.3), while achieving much higher recognition accuracy. These results highlight the importance of complete state observation for action recognition and also suggest that video generation performance can potentially reflect task difficulty.

**Choice of UNet layers.** As described in Section 3, the UNet mapping function $F$ is decoupled into $F_{tail} \cdot F_{head}$, where $F_{head}$ serves as the feature extractor for video recognition. Our UNet model contains 4 main up-sampling blocks. We investigate which one is best suited for recognition. As shown in Table 4, using the second up-sampling block (Up Index 2) yields the best performance with an accuracy of 75.8%. The third block (Up Index 3) followed with 75.2%, while the first block (Up Index 1) has the lowest accuracy. As such, we choose the second block for feature extraction.

**Explore the influence of the masking strategy.** We also conduct an ablation study on the masking schemes using different expected masking ratios, as shown in Table 5. The results show that the FVD scores remain similar across different ratios, and a larger masking ratio might be beneficial for generation, as it closely resembles our class-conditioned frame prediction scenario with one or two given frames. However, an excessively large masking ratio (87.5%) negatively impacts action recognition accuracy, leading to a 0.9% decrease compared to our selected ratio.

**Noise incorporation during inference for video recognition.** In the inference stage for action recognition, GenRec applies a specific level of noise to video inputs before extracting visual features, as formulated in Equation 18. This added noise helps maintain consistency with the noisy training

Table 6: SSV2 action recognition accuracy with different random seeds.

| Seed | 0 | 1 | 2 | 3 | 4 | 5 | AVG |
|------|---|---|---|---|---|---|-----|
| SSV2 Acc (%) | 75.83 | 75.82 | 75.83 | 75.83 | 75.86 | 75.84 | $75.835 \pm 0.0125$ |

process. To further understand the influence of noise randomness on classification accuracy, we conducted an experiment on the SSV2 dataset, using multiple random seeds to generate the noise, as presented in Table 6. The results demonstrate remarkable consistency in accuracy across different random seeds, with a standard deviation of only 0.0125%. This minimal variation highlights the model's robustness to noise fluctuations during inference, suggesting that the model's performance remains stable despite noise introduced by random sampling. If fully deterministic outcomes are desired, fixing the random seed for noise sampling will eliminate any remaining variability and guarantee consistent predictions across runs.

# 5 Related Work

**Video diffusion models for generation.**    The great success of diffusion models in image generation has led to rapid advancements in video generation, including text-to-video generation [15, 2, 52, 54], image&text-to-video generation [56, 18, 21], and video editing [4, 5, 26, 29, 12, 53]. Many current works [52, 18] adapts the diffusion models from images to videos by incorporating temporal convolutions and attention mechanisms. One typical and excellent work, Stable Video Diffusion [1], follows the above description and has provided valuable foundations for generating high-quality, diverse, and temporally consistent videos. Different from the previous work, in our paper, we pursue not only the quality of generation, but also the unity of model generation capability and classification ability.

**Diffusion models for visual understanding.**    Recently, researchers start to uncover the significance of diffusion models for discrimination tasks. A notable approach involves utilizing pretrained visual diffusion models for various downstream tasks, such as image segmentation [55] and visual content correspondence [39]. Additionally, some studies treat diffusion learning as a self-supervised method to acquire valuable feature representations [8]. However, most current works either use stable diffusion networks as pretrained backbones for downstream tasks or completely destroy their generative capabilities. Consequently, the potential benefits of integrating generation and classification abilities into a single model remain under-explored, which is the primary focus of our paper.

# 6 Conclusion

In this work, we presented GenRec, a unified video diffusion model that enables joint optimization for both video generation and recognition. GenRec exploits the significant temporal modeling power embedded in the diffusion model, allowing for mutual reinforcement between generation and recognition tasks. Extensive experiments were conducted to evaluate the performance of GenRec, demonstrate our approach contains strong generation and recognition capabilities at the same time in different kinds of scenarios, including normal or partial video recognition, video completion and class-conditioned image-to-video generation. Our findings highlight the potential of combining generation and classification tasks within a single unified model, providing valuable insights into the development of more sophisticated and versatile video analysis models. Future work will focus on further refining this integration and exploring its applications across various real-world scenarios.

# Acknowledgement

This work was supported in part by National Natural Science Foundation of China (#62032006). The authors would like to thank Rui Wang, Rong Bao, Rui Tian for their help and suggestions.

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

# A Appendix / supplemental material

## A.1 Early Prediction on EK-100 and UCF-101

More detailed evaluation results of the video recognition with limited frames on EK-100 and UCF-101 can be seen here.

Table 7: Early action prediction on EK-100.

| Method | Verb Obs. Ratio($\rho$) | | | | | Noun Obs. Ratio($\rho$) | | | | | Action Obs. Ratio($\rho$) | | | | |
|---|---|---|---|---|---|---|---|---|---|---|---|---|---|---|---|
| | 0.1 | 0.3 | 0.5 | 0.7 | 0.9 | 0.1 | 0.3 | 0.5 | 0.7 | 0.9 | 0.1 | 0.3 | 0.5 | 0.7 | 0.9 |
| TemPr [38] | 21.4 | 34.6 | 54.2 | 63.8 | 67.0 | 22.8 | 32.3 | 43.4 | 49.2 | 53.5 | 7.4 | 15.4 | 28.9 | 37.3 | 40.8 |
| MVD† [44] | 49.3 | 60.0 | 64.7 | 68.8 | 71.3 | 35.7 | 44.7 | 49.1 | 53.4 | 55.2 | 22.5 | 32.2 | 37.5 | 42.3 | 44.5 |
| GenRec | **55.8** | **63.6** | **67.9** | **71.7** | **73.1** | **40.1** | **47.8** | **52.0** | **55.3** | **56.7** | **28.1** | **36.1** | **40.8** | **45.0** | **46.6** |

Table 8: Early action prediction on UCF dataset.

| Method | Observation Ratio($\rho$) | | | | | | | | |
|---|---|---|---|---|---|---|---|---|---|
| | 0.1 | 0.2 | 0.3 | 0.4 | 0.5 | 0.6 | 0.7 | 0.8 | 0.9 |
| TemPr [38] | 88.6 | 93.5 | 94.9 | 94.9 | 95.4 | 95.2 | 95.3 | 96.6 | 96.2 |
| MVD† [44] | 90.8 | 94.2 | 94.8 | 95.7 | 95.5 | 96.2 | 96.5 | 96.8 | 96.9 |
| GenRec | **94.4** | **96.0** | **96.5** | **96.6** | **96.9** | **97.4** | **97.8** | **97.9** | **97.9** |

## A.2 Training and Inference Time Cost Compared with Previous Classification Methods

We compare the training and inference time cost between MVD-H, Baseline II and GenRec, fairly on the same hardware resource: 4-nodes * 8 V100s, and same batch size. Baseline II refers to fully fine-tuning the original SVD model with classification supervision only.

As shown in the Table 9, GenRec and the Baseline II consumes more testing time than MVD-H. The difference primarily arises from the varying number of parameters. Our model is derived from a generative model, and the complexity of the generative task necessitates a larger number of parameters for effective learning. Compared with Baseline II, since GenRec requires additional decoder blocks for video generation training, the training time will get increased a little bit. As the additional up-sampling blocks will not be used when doing action recognition, GenRec shares the same testing time with Baseline II. It is worth noting that "†" in MVD-H is to highlight that MVD method would use repeated augmentation technique during training, but such augmentation will significantly increase the training time. Our approach does not need to use that augmentation. All the methods do the down-stream finetuning for 30 epochs.

Table 9: Comparison with recognition methods in terms of parameters, training time, and test time.

| Method | Trainable Params | Total Params | Training Time | Training Epochs | Test Time (2 × 3 clips) |
|---|---|---|---|---|---|
| MVD-H† | 633M | 633M | 3038 s/Epoch | 30 | 441 s |
| Baseline II | 1.3B | 1.9B | 2954 s/Epoch | 30 | 1500 s |
| GenRec | 1.5B | 2.1B | 3751 s/Epoch | 30 | 1500 s |

## A.3 Ablation on Loss Balance Ratio

We conduct an ablation study on the loss balance ratio $\lambda$ as shown in Equation 9. Results presented in Table 10 show that varying the loss ratio has a minor impact on action recognition accuracy. However, setting the ratio too low negatively affects the generation performance. In particular, when the ratio is set to zero, significant forgetting occurs, severely compromising the model's generation ability.

Table 10: Ablation study on the effect of different loss balance ratios on SSv2 accuracy and FVD. Higher SSV2 accuracy and lower FVD indicate better performance.

| Balance Ratio $\lambda$ | 0 | 1 | 5 | 10 | 20 |
|---|---|---|---|---|---|
| SSv2 Acc ↑ | 75.6 | 75.5 | 75.6 | 75.8 | 75.5 |
| SSv2 FVD ↓ | 1579.2 | 52.0 | 47.4 | 46.5 | 46.9 |

## A.4 Comparision with SVD Baseline

Comparing with the open-source Stable Video Diffusion directly is meaningful. We conduct frame prediction tests using SVD model on the SSV2 and Epic-Kitchen datasets, as shown in Table 11. Since the SVD model performs better at higher resolutions, we generated videos at 512x512 resolution and then downsampled them to 256x256 for FVD calculations. The results show that while SVD achieves competitive scores compared to previous state-of-the-art methods shown in Table 1, it is suboptimal compared to GenRec. This is likely due to SVD's design for general scenarios. Baseline I represents our enhanced SVD baseline, which fine-tunes SVD on the target datasets for better results and fairer comparison with GenRec.

Table 11: Comparison of SSv2 FVD and Epic FVD scores in frame prediction tests.

| Method | SSv2 FVD | Epic FVD |
|---|---|---|
| **SVD** | **99.7** | **180.8** |
| Baseline I | 50.3 | 53.6 |
| GenRec | 46.5 | 49.3 |

## A.5 Impact of Classification Training on Generative Ability

To demonstrate classification and generative can coexist well in our training without negatively impacting each other, we construct the comparison with "Baseline I (SVD baseline)", which adapts SVD on the target dataset without classification supervision, and then trains another classifier head freeze the generation backbone, ensuring no influence from classification supervision. We conduct comparisons on SSv2: FP (frame prediction) and CFP (class-condition frame prediction), as shown in Table 12. By comparing the FP results, the similar scores between the two methods conclude that classification loss does not degrade the model's generative ability in our approach. Moreover, the CFP results indicate that our method, with its more accurate classification performance, can guide the model to achieve higher-quality video frame predictions.

Table 12: Comparison of SSv2 Acc and SSv2 FVD scores for Baseline I and GenRec methods in FP (frame prediction) and CFP (class-condition frame prediction) scenarios.

| Method | SSv2 Acc ↑ | SSv2 FVD ↓ |
|---|---|---|
| Baseline I (FP) | - | 55.5 |
| GenRec (FP) | - | 55.3 |
| Baseline I (CFP) | 63.7 | 50.3 |
| GenRec (CFP) | 75.8 | 46.5 |

## A.6 Case Study for Class-Conditioned Image-to-Video Generation and Video Interpolation

We show the generated visualization of the GenRec. The model can support video generation given various numbers of frames, as well as category-guided generation. We show two of the most difficult generative scenarios, which are: (1) given the first frame and different action categories to guide the video generation, and (2) given the start and end frames, the model is expected to complement the video. We also compare our methods with SEER [18] with cases picked from its official website. The generation results can be seen in Figure 4, Figure 5 and Figure 6.

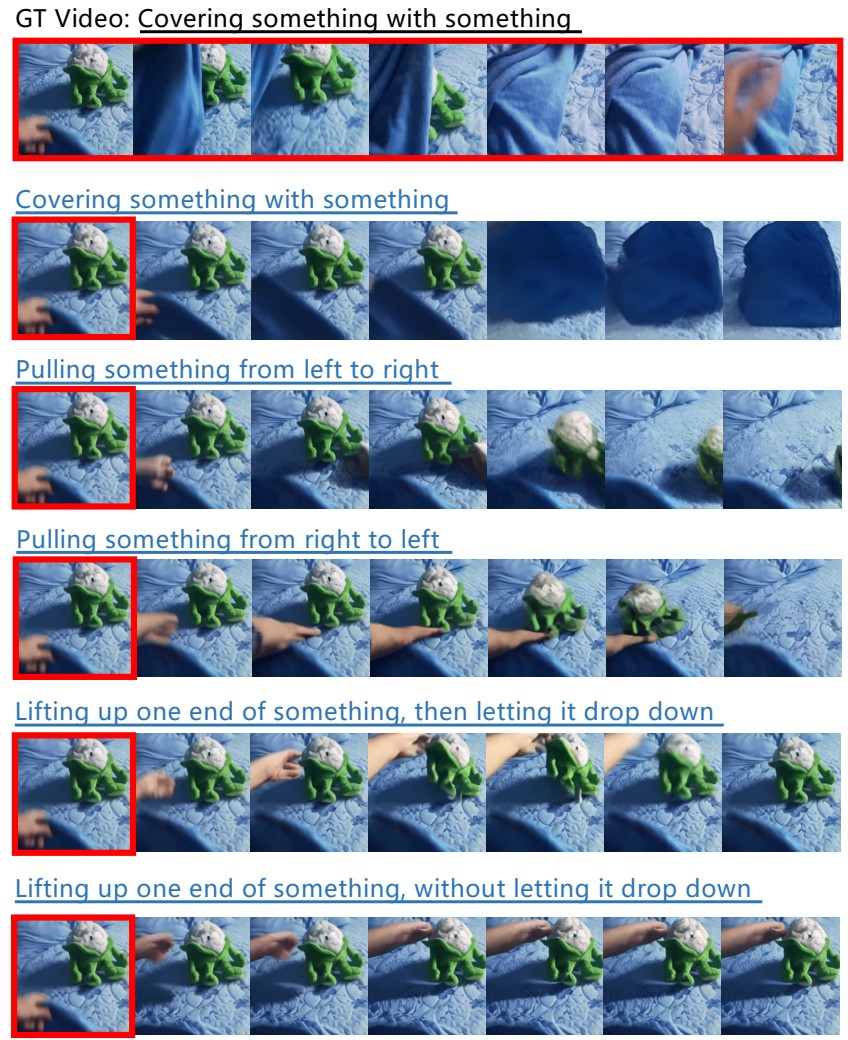

Figure 4: Video generation case study. We generate videos given the first frame together with the classifier guidance for various categories.

## A.7 Limitations and Broader Impacts

**Limitations and Future Work**  The objective of our paper is to unify the tasks of generation and recognition, achieving or even surpassing the state-of-the-art experimental performance across various tasks. However, our method is based on fine-tuning a pretrained video diffusion model, using more pretraining data and having a larger number of parameters compared to previous methods. This is an issue we need to address in the future, and exploring the distillation of a well-pretrained video diffusion model into a smaller model is a worthwhile future endeavor.

**Broader Impacts**  The broader impact of the GenRec framework extends into various fields, enhancing capabilities in content creation, security, and accessibility. In the media industry, it allows for the automated generation of tailored, high-quality videos, reducing production costs and fostering creativity. For surveillance, its robustness in limited information scenarios improves monitoring effectiveness, particularly in challenging environments. Additionally, advancements of GenRec in video prediction can aid in developing assistive technologies, making digital content more accessible and interactive, particularly for individuals with visual impairments.

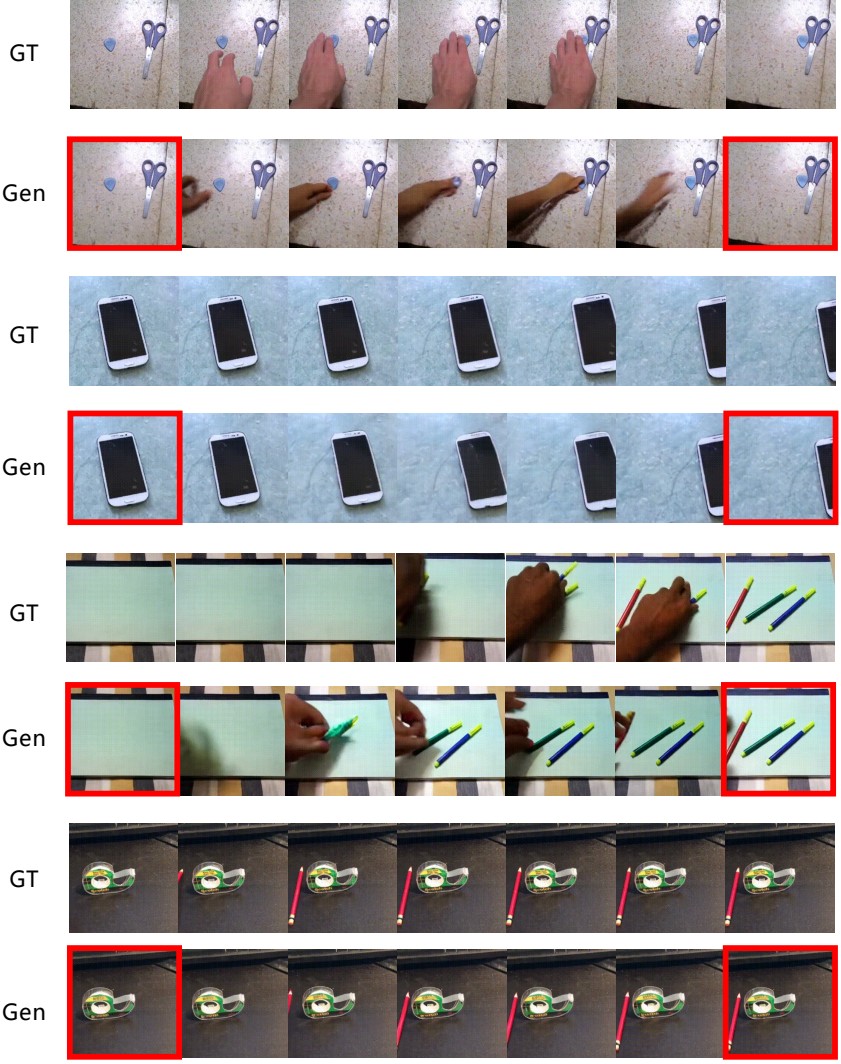

Figure 5: Video generation case study. We generate videos given the first frame and the last frame.

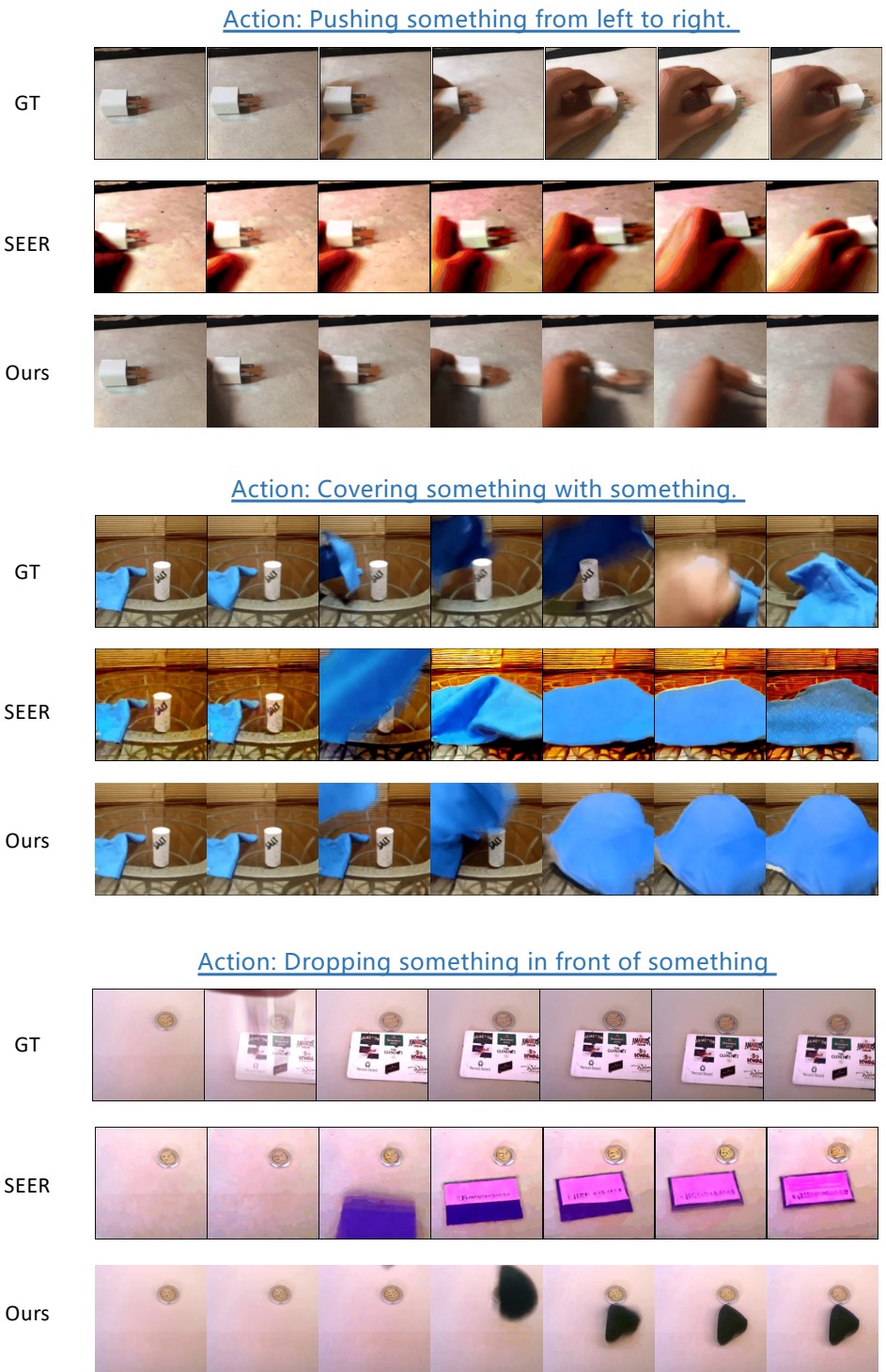

Figure 6: Video generation case study. We compare our methods with SEER [18] in the setting of generating videos given the first frame together with the classifier guidance. Cases are picked from the official website of SEER [18].

