# OpenReview forum: "GenRec: Unifying Video Generation and Recognition with Diffusion Models"
_NeurIPS.cc/2024/Conference — NeurIPS 2024 poster_

### Official Review · Reviewer_C3Pd · 2024-07-08

**Soundness:** 3
**Presentation:** 2
**Contribution:** 3
**Rating:** 5
**Confidence:** 5

**Summary:**

This paper explores the use of video diffusion models for joint generation and recognition. The authors introduce a framework called GenRec, based on Stable Video Diffusion, that aims to unify video generation and recognition. GenRec employs a random-frame conditioning process to learn generalized spatial-temporal representations, making it robust even with limited visual inputs. Experiments show that GenRec excels in both video recognition and generation. Additionally, GenRec proves to be highly robust in scenarios with limited frame availability.

**Strengths:**

1. The integration of MAE and Diffusion models is a practical and innovative approach to unify video generation and recognition tasks.

2. The experiments are comprehensive, covering video generation, video prediction, and video recognition. The provided results demonstrate strong performance of the model across all tasks.

**Weaknesses:**

1. The notations used in the Preliminary Section are inconsistent. The authors should review and correct the formulations for clarity.

2. In Section 3.3, the authors claim to have adopted the stochastic sampler in EDM, but it appears that the second-order correction is not included according to Algorithm 2 in EDM. The authors are encouraged to provide justifications for this omission.

3. In the derivation on Line 156, it is unclear why such a replacement can be used for the original score function. Additionally, the derivation from Equations 14 to 16 lacks clarity. The authors should provide more detailed explanations.

4. In Table 1, the generative performance of Baseline I should be included. Furthermore, information on parameter counts and computational overhead should also be presented.

5. There are misspellings such as "mensioned" on Line 135 and "interatively" on Line 147. These should be corrected.

**Questions:**

Can you provide more details on the derivation of Eq.14-16?

**Limitations:**

The authors have mentioned the limitations.

---

> ### Author Rebuttal · Authors · 2024-08-06
>
> Thank you for your valuable comments. We will address your concerns point by point.
>
> **Q1: Provide more details on the derivation of Eq.14-16.**
>
> The derivation of Eqs. 14-16 is similar to Eq. 12 in **[A]**. Here are the details.
>
> Firstly, Eq. 13 in our paper shows the formulation of the unconditional noise reduction residual.
> $$
> \epsilon_{\theta}(\hat{z_i}; \overline{z_0})=(-1)(t_{i-1}-\hat{t_i}) \nabla_{\hat{z_i}} \log p_{\theta} (\hat{z_i})
> $$
> To introduce the class control, we substitute the score function $\nabla p(\hat{z_i})$ with the conditional form $\nabla_{\hat{z_i}} \log p_{\theta} (\hat{z_i}|y)$, aiming it to be the conditional distribution we want. It is worth noting that the substitution introduces class information and is not an equivalent replacement. Next, by applying the Bayes` Theorem, the score function becomes:
>
> $$
> \nabla_{\hat{z_i}} \log p (\hat{z_i}|y)=\nabla_{\hat{z_i}} \log \frac{p (y | \hat{z_i}) \cdot p(\hat{z_i})   }{p(y)}
> =\nabla_{\hat{z_i}} \log p(y | \hat{z_i}) + \nabla_{\hat{z_i}} \log p(\hat{z_i}) - \nabla_{\hat{z_i}} \log p(y)
> =\nabla_{\hat{z_i}} \log p(y | \hat{z_i}) + \nabla_{\hat{z_i}} \log p(\hat{z_i})
> $$
>
> Evidently, ${\nabla_{\hat{\mathbf{z}}_i}} \log p(y)=0$ and can be removed. As a result, the conditional noise reduction residual formulation will become,
>
> $$
> \epsilon_{\theta}^{*}(\hat{z_i}; \overline{z_0})=(-1)(t_{i-1}-\hat{t_i}) \nabla_{\hat{z_i}} [\nabla_{\hat{z_i}} \log p(y | \hat{z_i}) + \nabla_{\hat{z_i}} \log p(\hat{z_i})]
> $$
> which corresponds to the Eq. 15 in our paper. We will update this section with the above detailed derivation process in the revised version.
>
> [A] Prafulla Dhariwal and Alexander Nichol. Diffusion models beat gans on image synthesis. In NeurIPS, 2021.
>
> **Q2: The authors are encouraged to provide justifications for the EDM second-order correction ommision**
>
> We follow the setting of SVD implementation, which uses the EulerDiscreteScheduler within the EDM framework, as can be seen in the **``huggingface, stable-video-diffusion-img2vid-xt, scheduler/scheduler\_config.json''**. EulerDiscreteScheduler do the denoising step without second-order correction, so in our previous version, we omit the second order correction.
>
> In addition, with the Eq. 17 that reveals the calculating way for the class-condition denoising residual  $\epsilon^{*}_{\theta}(\hat{\mathbf{z}}_i;\overline{\mathbf{\mathbf{z}}_0})$,
>
> we can substitute it into the algorithm 2 in EDM to get the second-order expression as below:
> $$
> \begin{align}
>     z_{i-1}^{corr}
>     & = z_i + \frac{1}{2}{(t_{i-1} - \hat{t_i})}(d_i + d_{i}^{'}) = z_i + \frac{1}{2} {\epsilon_{\theta}^{\*} (\hat{z_i};\overline{z_0})} + \frac{1}{2}\frac{t_{i-1} - \hat{t_i}}{t_{i-2}-t_{i-1}} \epsilon_{\theta} ^{\*} (\hat{z_i} + {\epsilon_{\theta}^{*} (\hat{z_i};\overline{z_0})};\overline{z_0})
> \end{align}
> $$
>
> We will add the above illustration and the derivation process in the final revised version.
>
> **Q3: In Table 1, the generative performance of Baseline I should be included. Furthermore, information on parameter counts and computational overhead should also be presented.**
>
> Thank you for your insightful suggestions. We acknowledge the importance of including the generative performance of Baseline I. We have supplemented Baseline I with its generation scores to act as the generation baseline, shown as bellow.
>
> | **Method** | SSv2 Acc  $\uparrow$ | K400 Acc  $\uparrow$ | SSv2 FVD  $\downarrow$ | Epic FVD  $\downarrow$ |
> | :--: | :--: | :--: | :--: | :--: |
> |Baseline I | 63.7 | 82.0  | 50.3 | 53.6 |
> |GenRec | 75.8 | 87.2 | 46.5 | 49.3 |
> ||
>
> Also the parameter counts information has been added in the **updated PDF, Table 1**.
>
> As for the computational cost/time analysis, we compare the training and recognition inference time cost between MVD-H, baseline II and GenRec, fairly on the same hardware resource: 4-nodes * 8 V100s, and same batch size. Baseline II refers to fully fine-tuning the original SVD model with classification supervision only.
> As shown in the table, both GenRec and Baseline II consume more test time than MVD-H, primarily due to the higher number of parameters. GenRec requires additional decoder blocks for video generation, which slightly increases the training time compared to Baseline II. However, these blocks are not used during action recognition, so GenRec and Baseline II share the same test time. Note that the ‘‘✝’’ in MVD-H indicates the use of repeated augmentation, which significantly increases the training time. Our approach doesn't need to use that augmentation. All methods fine-tune downstream for 30 epochs.
>
> |**Method** | Trainable Params | Total Params | Train Time | Train epochs | Test Time (2$\times$3 clips) |
> |:--:|:--:|:--:|:--:|:--:|:--:|
> |MVD-H ✝ |  633M | 633M  | 3038 s/Epoch| 30 | 441s |
> |Baseline II | 1.3B | 1.9B | 2954 s/Epoch | 30 | 1500s |
> | GenRec | 1.5B | 2.1B | 3751 s/Epoch | 30 | 1500s |
>
> The above discussions will be included in our revised version.
>
> **Q4: misspelling words and notation inconsistent**
>
> Thanks for your carefully reading. We will definitely check and revise, and make sure of the consistency in the revised version.

---

> ### Comment · Reviewer_C3Pd · 2024-08-13
> **Post-rebuttal comments**
>
> Many thanks for the authors' response. I've checked the authors' rebuttal and other reviewers' comments. I think the response has addressed my concerns. Therefore, I decide to raise the score to 5. Moreover, is it possible for the authors to provide the derivation from Eq. 3 to Eq. 4? The derivation is a bit confusing to me.

---

> > ### Author Response · Authors · 2024-08-13
> >
> > Thank you for reviewing our work and providing valuable feedback. We are pleased that we have been able to address your concerns.
> >
> > - For the Eq. 3 and Eq. 4 in the preliminary section of our paper, the equations are from Eq. 2 and Eq. 3 in the EDM paper.
> >
> > - For the Eq. 3 to Eq. 4 in our rebuttal, here is a detailed explanation:
> >
> > Eq. 17 in our paper outlines the calculation method for the class-conditioned denoising residual $\epsilon^{*}_{\theta}(\hat{\mathbf{z}}_i;\overline{\mathbf{z}}_0)$. We can apply this iteratively as follows:
> >
> > $$
> > z_{i-1} = \hat{z_i} + \epsilon_{\theta}^{*}( \hat{z_i};\overline{z_0} )
> > $$
> >
> > $$
> > z_{i-2}= {z_{i-1}}  + \epsilon_{\theta}^{*}(z_{i-1};\overline{z_0})
> > $$
> >
> > Following line 8 in Algorithm 2 of EDM, we can compute the terms:
> >
> > $$
> >     d_{i} = \frac{{z_{i-1}} - \hat{z_{i}}}{t_{i-1}-\hat{t_{i}}} = \frac{ \epsilon_{\theta}^{*}( \hat{z_i};\overline{z_0})  }{t_{i-1}-\hat{t_{i}}}
> > $$
> >
> > $$
> > \begin{align}
> >     d_{i}^{'}  = \frac{z_{i-2}-z_{i-1}}{t_{i-2}-t_{i-1}} = \frac{\epsilon_{\theta}^{\*}(z_{i-1};\overline{z_0}) }{t_{i-2}-t_{i-1}}
> > =\frac{\epsilon_{\theta} ^{\*} (\hat{z_i} + {\epsilon_{\theta}^{\*} (\hat{z_i};\overline{z_0})};\overline{z_0})  }{t_{i-2}-t_{i-1}}
> > \end{align}
> > $$
> >
> > Using the second-order correction, and by substituting these two terms into line 11 of Algorithm 2 in EDM, we obtain:
> >
> > $$
> > \begin{align}
> > z_{i-1}^{corr} & = z_i + \frac{1}{2}{(t_{i-1} - \hat{t_i})}(d_i + d_{i}^{'}) = z_i + \frac{1}{2} {\epsilon_{\theta}^{\*} (\hat{z_i};\overline{z_0})} + \frac{1}{2}\frac{t_{i-1} - \hat{t_i}}{t_{i-2}-t_{i-1}} \epsilon_{\theta} ^{\*} (\hat{z_i} + {\epsilon_{\theta}^{*} (\hat{z_i};\overline{z_0})};\overline{z_0})
> > \end{align}
> > $$
> >
> > Thank you again for your positive recognition and thoughtful feedback. We greatly appreciate the time and effort you have taken to review our work.

---

### Official Review · Reviewer_96MS · 2024-07-11

**Soundness:** 4
**Presentation:** 4
**Contribution:** 4
**Rating:** 9
**Confidence:** 5

**Summary:**

The paper proposes to unify video recognition and generation with a masked finetuning of SVD (diffusion video model). By training with both classification and generation objectives, the model learns to do both. Rather than training with first frame only for conditioning, the masking is employed so that the classifier can learn better. The resulting model has recognition performance that is actually enhanced by the generative part of the training, and results are given across various tasks and subtasks, including action prediction and generating videos given multi-frame conditioning.

**Strengths:**

[S1] The method has good results as a unified model, and introduces a novel way to finetune video diffusion for both recognition and generation, performing very well on both tasks. Such a system is important because ideally it saves on storage cost (no need to store a model for each task, just store 1 for both), training time, makes better use of data (training with supervised text losses and unsupervised generative losses), and should synergize between the tasks to form a true world model.

[S2] The fact that the generative training helps the classification performance, as shown in Table for example, offers critical evidence to support the paper's claim that it unifies generation and recognition (rather than them competing with each other).

[S3] Baseline I and Baseline II are very helpful for contextualizing the performance of the model, showing what is possible with the CE training alone.

**Weaknesses:**

[W1] It is unclear why the paper compares with methods that pretrain on only kinetics and imagenet (like Hiera) when GenRec uses the SVD which is pretrained on massive amounts of external data. For the recognition task, appropriate SOTAs would be OmniVec, OmniVec2, InternVideo, InternVideo2, etc.

[W2] FVD is not a very good metric. A generative paper ought to have generative results in the main paper itself, FVD results alone are not adequate, and such examples ought not be saved for supplementary. Additionally, the examples in the supplementary are not adequate, since there is no comparison to the competing methods.

[W3] Paper does not cite enough related work. For example, [1] and [2] are omitted, as are many others that explore similar diffusion-for-recognition directinos.

[1] Your Diffusion Model is Secretly a Zero-Shot Classifier, Li et al.
[2] Diffusion Models Beat GANs on Image Classification, Mukhopadhyay et al.

**Questions:**

How did you select your SOTAs? One of the main things preventing me from giving a higher rating is that the model does considerably lag behind SOTAs for k400 classification.

Why do you not provide example videos for the competing methods, as reference points? It is difficult to assess the capabilities of the video generation based on FVD alone.

Why is there no generative baseline, similar to the classification baselines?

**Limitations:**

Yes.

---

> ### Author Rebuttal · Authors · 2024-08-06
>
> Thanks for the positive feedback! We address your questions below.
>
> **Q1: SOTA selection and completion.**
>
> We greatly appreciate your suggestion. We agree that including more comparable methods is very meaningful. **We have updated the Table 1 in the uploaded PDF** to include more comparable methods, specifically InternVid, InternVid2, OmniVec, and OmniVec2, and categorized them into single modality and multi-modality for clearer comparison.
>
> Our initial version primarily focused on comparing with single-modality self-supervised methods. While the newly added methods incorporate multiple modalities, introducing multiple modality encoders for multi-modal alignment on large datasets, building upon previous self-supervised methods. Although SVD is a model that is pretrained on additional data, it does not explicitly perform multi-modal alignment during the generation training process, making it hard to surpass their performance. Indeed, unifying the generation task and visual-language multi-modal alignment is a promising research direction and can be achieved by extending our work.
>
> **Q2: Provide example videos for the competing methods, as reference points.**
>
> Thanks for your helpful suggestions. We have added more detailed example comparison with competing methods, as shown in the **uploaded PDF, Figure 1 and Figure 2**. In the update, we compare examples provided by SEER official website. Also we add more comparison among the closed-source generation methods (PIKA, Gen-2, KLING) and open-source method (SEER) on action control cases. From the cases in the PDF, we find that:
> SEER suffers from distortion issues and lacks accuracy, e.g, easy to get wrong for the left-right direction control. PIKA maintains strong visual consistency but fails to generate accurately according to instructions. Gen-2 often has problems with object deformation. KLING shows very high visual consistency, and for some instructions it can generate high-quality correct videos, e.g, cases of ``covering something<the toy> with something<a blue towel>''. But sometimes it struggles with fine-grained action control. Our method demonstrates good performance, with higher visual quality and more accurate fine-grained action control, compared with other methods.
>
> More comprehensive cases comparison will be added in our final revision version.
>
> **Q3: Need Generation baseline.**
>
> Thank you for your insightful suggestions. We acknowledge the importance of including a generation baseline for SVD. Baseline I can serve as a robust generation baseline, and we have supplemented it with its generation scores to act as the generation baseline, shown as bellow table. We would update this in our revised paper version.
>
> | **Method** | SSv2 Acc  $\uparrow$ | K400 Acc  $\uparrow$ | SSv2 FVD  $\downarrow$ | Epic FVD  $\downarrow$ |
> | :--: | :--: | :--: | :--: | :--: |
> |Baseline I | 63.7 | 82.0  | 50.3 | 53.6 |
> |GenRec | 75.8 | 87.2 | 46.5 | 49.3 |
> ||
>
> **Q4: Paper should cite more enough works**
>
>  Thanks. We would cite your mentioned related works in the revised version.

---

> > ### Comment · Reviewer_96MS · 2024-08-12
> > **One more question**
> >
> > With these clarifications, I am inclined to increase my rating.
> >
> > One quick question- what is OmniViD in table 1 of the rebuttal?

---

> > > ### Author Response · Authors · 2024-08-12
> > >
> > > Thank you so much. We apologize for the naming mistake in Table 1 of the rebuttal which should be "GenRec".
> > >
> > > Thank you again for your recognition and thoughtful feedback. We greatly appreciate your time and effort in reviewing our work.

---

### Official Review · Reviewer_JBof · 2024-07-13

**Soundness:** 3
**Presentation:** 2
**Contribution:** 2
**Rating:** 5
**Confidence:** 3

**Summary:**

This paper investigates to unify the video generation and recognition tasks into the same diffusion model. The authors propose a conditional feature mask mechanism to unify the two learning tasks. Experiments are conducted on several tasks to validate the effectiveness of the proposed method.

**Strengths:**

The idea of unifying generation and recognition process into the same diffusion model is interesting and necessary for the computer vision.
The modification to the SVD backbone is simple, including a masked conditional input and a classifier head.

**Weaknesses:**

1. The design of unifying generation and recognition tasks is not elegant. This paper directly introduces an extra classification head, and uses a multi-task learning loss for these two tasks. As verified in the Table 4, the trade-off between high-level semantic recognition task and low-level detail synthesis is not controlled well and may be conflict.
2. The experimental results are not promising. Firstly, the model size and computational cost should be reported in Table 1. It will be clear to analyze the efficiency of the recognition model. Secondly, for the generation ability evaluation, since the model is initialized from the SVD, it’s better to compare with SVD on different datasets. A concern is that whether the classification loss training will degrade the model’s synthesis ability on natural scene. It’s over claim to say “surpassing the sota”.
3. The mask mechanism in the proposed method shares some similarity with the paper [1]. The authors are suggested to add a discussion with this work.
4. For the recognition task, the proposed method just uses the SVD backbone to extract features, since z0 is directly used. Such a design (i.e., extracting features from the input video and sending them to a classifier)  is conventional.


[1] Masked Diffusion Transformer is a Strong Image Synthesizer, ICCV 2023.

**Questions:**

1. Why the loss weight of the recognition loss in Eq. 8 is not related with the noise level?
2.  How about the training and testing efficiency compared with other recognition models?

---

> ### Author Rebuttal · Authors · 2024-08-07
>
> Thank you for your valuable comments. We will address your concerns point by point.
>
> **Q1: Why the loss weight of the recognition loss is not related with the noise level?**
>
> Thanks for your valuable feedback. Adding loss weighting related to noise is standard for diffusion training, but is unexplored for classification. Designing the dynamic classification loss weight brings additional hyperparameters, influenced by factors such as noise level and masking ratio. Note that in EDM generation, small loss ratio are set when the noise levels are large and small. However, for classification, we aim for robust partially video recognition with varing noise. Thus, we maintain a constant classification loss weight to ensure stability, and also to reduce the number of hyperparameters, making training more manageable. Experimental results demonstrate our design effectively balances both generation and classification.
>
> **Q2: Training and testing efficiency/computational cost, compared with other recognition models.**
>
> Thanks for your valuable feedback. We compare the training and inference time costs between MVD-H, Baseline II, and GenRec, using the same hardware (4 nodes * 8 V100s) and batch size. Baseline II refers to fully fine-tuning the SVD with classification only.
>
> As shown in the table, both GenRec and Baseline II consume more test time than MVD-H, primarily due to the higher number of parameters. GenRec requires additional decoder blocks for video generation, which slightly increases the training time compared to Baseline II. However, these blocks are not used during action recognition, so GenRec and Baseline II share the same test time. Note that the ‘‘✝’’ in MVD-H indicates the use of repeated augmentation, which significantly increases the training time. Our approach doesn't need to use that augmentation. All methods fine-tune downstream for 30 epochs.
>
> |**Method**|Trainable Params|Total Params|Train Time|Train epochs|Test Time (2$\times$3 clips)|
> |:-:|:-:|:-:|:-:|:-:|:-:|
> |MVD-H ✝|633M|633M|3038 s/Epoch|30|441s|
> |Baseline II|1.3B|1.9B|2954 s/Epoch|30|1500s|
> |GenRec|1.5B|2.1B|3751 s/Epoch|30|1500s|
> ||
>
> **Q3: Model size.**
>
> Thanks. We have updated Table 1 in the uploaded PDF to include parameter counts.
>
> **Q4: Concern about whether the classification training will degrade the model generation ability.**
>
> Thank you for your valuable feedback. To demonstrate classification and generative can coexist well in our training without negatively impacting each other, we have constructed the comparison with "Baseline I (SVD baseline)", which adapts SVD on the target dataset without classification supervision, and then trains another classifier head freeze the generation backbone, ensuring no influence from classification supervision. We conduct comparisons on SSv2: FP (predicting future frames) and CFP (predicting future frames with class information).
>
> By comparing the FP results, the similar scores between the two methods conclude that classification loss does not degrade the model’s generative ability in our approach. Moreover, the CFP results indicate that our method, with its more accurate classification performance, can guide the model to achieve higher-quality video frame predictions.
>
> We will update the above discussion in our revised version.
>
> |**Method**|**SSv2 Acc $\uparrow$**|**SSv2 FVD $\downarrow$**|
> |:-:|:-:|:-:|
> |Baseline I (FP)|-|55.5|
> |GenRec (FP)|-|55.3|
> ||
> |Baseline I (CFP)|63.7|50.3|
> |GenRec (CFP)|75.8 |46.5|
> ||
>
> **Q5: Mask mechanism discussion with [A].**
>
> Thank you for your valuable suggestions. In [A], the authors choose to mask on the noisy latents directly, while we choose to mask the condition latents. Masking noisy latents can break the mathematical theorem basis and complicate the task. Therefore, we opt not to influence the latent noise but to change the conditional latents instead.
>
> We will add the discussion about [A] in the related work section of our revised version.
>
> [A] Masked Diffusion Transformer is a Strong Image Synthesizer, ICCV 2023.
>
> **Q6: Table 4 results show recognition and generation may be conflict.**
>
> Thanks. Table 4 presents an ablation study on the layer index for feature extraction, showing proper layer selection is needed for preventing conflict. Shallow feature extraction decreases performance due to insufficient complexity, while late feature extraction conflicts with generation. Thus, we selected an optimal layer to balance performance and prevent conflicts between generation and understanding tasks.
>
> For further demonstration, we **upload the PDF** and compare with baseline performance. In the table 1, Baseline I train the diffusion process without classification, and Baseline II trains classification without generation. The results show our method outperforms Baseline I in generation and matches or exceeds Baseline II in classification. These findings confirm our final GenRec model does not exhibit conflicts between generation and classification.
>
> **Q7: SVD to extract features is conventional, and the design of unifying generation and recognition is not elegant.**
>
> Thanks. We would like to stress that applying diffusion models for video understanding is non-trivial due to the complicated nature of videos. Moreover, unifying generation and recognition through diffusion models is even more challenging, as conflicts can easily arise and mutual enhancement between the two tasks is difficult to achieve. **This is precisely where our GenRec stands out.**
>
> By exploring the temporal modeling capabilities of SVD and integrating our novel random masking condition, GenRe effectively bridges the gap between generation and classification, and achieves significant mutual enhancement between the two types of tasks. The effectiveness confirms our design is innovative and pragmatic, and we believe GenRec represents an exciting direction for the computer vision community by combining generation and recognition tasks in a unified framework.

---

> > ### Author Response · Authors · 2024-08-14
> >
> > Dear Reviewer JBof,
> >
> > Hope this message finds you well. We wanted to kindly follow up on the feedback for our submission. Your insights are invaluable, and we would greatly appreciate any further comments you may have.
> >
> > Thank you for your time and consideration.

---

### Official Review · Reviewer_PRtT · 2024-07-14

**Soundness:** 3
**Presentation:** 4
**Contribution:** 3
**Rating:** 7
**Confidence:** 4

**Summary:**

Recently, diffusion models have been released that can generate high quality video clips, indicating that they strongly capture natural appearance and motion in video. This paper seeks to adapt one recent model, Stable Video Diffusion (SVD), to explore if what it has learned can be leveraged to improve video recognition performance, and investigate whether co-training on generation and recognition using this model can improve on both tasks.

The two tasks, video diffusion and recognition, have quite different setups: the first takes uncorrupted video as input, while the second starts with a single conditioning frame as well as noise-corrupted video. This paper proposes to unify the tasks, adapting the SVD model to take masked-uncorrupted video plus noised video, concatenated, as input. This unifies the tasks, allowing joint training on recognition and generation. It also permits new tasks: generation and recognition using a variety of partially-masked video configurations as input.

The paper shows competitive results on classic video clip classification tasks, and very strong results on generation FVD. Further they demonstrate strong classification performance on masked input scenarios including early action prediction, and recognition with limited frame-rates.

**Strengths:**

This paper addresses what is to me, a researcher in video recognition and generation, a really burning question: generative video models have clearly learned an incredible amount about the visual world, so how can we harness that to advance recognition performance?

But the setups of the problems, recognition and generation via diffusion, are really quite different. The authors show how to unify them, and co-train on the two tasks, in an original way that actually produces synergies for both. This is an exciting result to me.

The paper is very clearly written and easy to follow (although the math dump in section 3.3 could be a little clear).

I appreciate the inclusion of the two strong baselines (I and II, described on line 213), that compare the unified model to models trained only on generative or only on discriminative tasks.

**Weaknesses:**

In the recognition results, Table 1 doesn't represent the current SOTA. E.g. InternVideo (https://arxiv.org/abs/2212.03191) from 2022 claims 77.2% on SSv2 and 91.1% on K400. It's fine if GenRec doesn't beat SOTA, but SOTA should be accurately represented in the table.

The generative experimental results show a surprising improvement on FVD on SSv2 and EK-100, which I don't feel the authors do enough to explain or analyze.  (See an associated "Question" below.)

There is no exploration of how different values of the generative/discriminative loss balance ratio, gamma, affects performance on the benchmarks.

Similarly, there is no exploration of different latent masking schemes or ratios during training.

Minor weaknesses:
* The method is described as "analogous to a temporal extension of MAE" in line 53. VideoMAE [38] does already exist.
* On line 61, please define the accuracy retention ratio of include a citation.

**Questions:**

The generative results in Table 1 demonstrate a really surprising decrease in FVD compared to the other methods. Can you explain why this is the case in more detail? Is the model really that good? Is it a typo? Is it because this benchmark is not particularly competitive? (I think FVD is more often presented on UCF and Kinetics-400 in the papers I've read, e.g. https://arxiv.org/pdf/2212.05199). If you double-check the FVD result, and convince me, then I'll gladly list this surprising decrease as one of the paper's strengths.

Would it be useful to clarify to readers why the focus is "particularly the unconditioned or image-conditioned models" in line 29?

**Limitations:**

Yes

---

> ### Author Rebuttal · Authors · 2024-08-06
>
> Thanks for the positive feedback! We address your questions below.
>
> **Q1:SoTA results completion.**
>
> Thank you for your valuable feedback. We have updated the Table 1 in the **uploaded PDF**, which includes more comparisons with methods like InternVid, InternVid2, OmniVec and OmniVec2. In addition, we have included the parameter counts for each method for a more comprehensive comparison.
>
> It is important to note that, our initial version primarily focused on comparing with single-modality self-supervised methods. In contrast, the newly added methods, e.g, InternVid2, incorporate more modalities, introducing multiple modality encoders for multi-modal alignment on large data, and is built upon the previous self-supervised methods. Therefore, we have distinguished between these two types of methods in the updated table.
>
> **Q2:More analysis and explanation about the FVD scores.**
>
> Thank you for your valuable feedback. We have double-checked our FVD results and confirmed their accuracy. The reduction in FVD can be attributed to two main factors: The use of the SVD model and the guidance of an accurate classifier. Among two of them, the application of the SVD pretrained model is the primary reason.
>
> To illustrate this point, we have added the FVD scores for baseline I for frame prediction and class-conditioned frame prediction. Results are shown as below. It is evident that using the fine-tuned SVD of "baseline I" for frame prediction (FP, given the first two frames and predicting the subsequent video frames) yields good scores as shown in the first line of the table below, thanks to the rich priors learned during the SVD pretraining on large-scale video data. In contrast, other methods primarily rely on image-based stable diffusion models and fine-tune them on limited video data, making it more challenging to learn temporal dynamics and resulting in lower performance. In addition, the table below also shows that precise classifier guidance, corresponding to CFP (class-condition frame prediction), leads to better FVD results.
>
> In summary, the major cause for the low FVD is the usage of pretrained SVD model, along with the significant impact of accurate classification guidance. We will include the analysis and explanation of the FVD scores in our revised version.
>
> | **Method** | **SSv2 Acc $\uparrow$** | **SSv2 FVD $\downarrow$** |
> | :--: | :--: | :--: |
> |Baseline I (FP) | - | 55.5 |
> |Baseline I (CFP) | 63.7 | 50.3 |
> |GenRec (CFP) | 75.8 | 46.5 |
> ||
>
> **Q3:Ablation on the loss balance ratio and the masking schemes.**
>
> Thank you for your valuable feedback. We conduct an ablation study on the loss balance ratio, and the results are presented below. The study shows that varying the loss ratio has a minor impact on action recognition accuracy. However, setting the ratio too low negatively affects the generation performance. In particular, when the ratio is set to zero, significant forgetting occurs, severely compromising the model’s ability to perform the generation task.
>
> | |  |  |  | |  |
> | :--: | :------: | :--: | :--: | :--: | :--: |
> | **Balance Ratio** |  0 | 1 | 5 | 10 | 20 |
> | **SSv2 Acc $\uparrow$**| 75.6 | 75.5 | 75.6 | 75.8 | 75.5 |
> | **SSv2 FVD $\downarrow$** | 1579.2 |52.0 | 47.4 | 46.5 | 46.9 |
> ||
>
> We also ablate the masking schemes with different masking ratio expectation. The results show that a larger masking ratio might be beneficial for generation, as it closely resembles our class-conditioned frame prediction with one or two given frames setting. However, too large masking ratio (87.5\%) harms the action recognition accuracy, resulting in a 0.9% decrease compared to our chosen ratio.
>
> These ablation results will be included in the final revised version.
>
> | **Expectation of Masking Ratio** | **SSv2 Acc $\uparrow$** | **SSv2 FVD $\downarrow$** |
> | :--: | :--: | :--: |
> |**75\% (our choice)** | 75.8 | 46.5 |
> |50\% | +0.3 | +0.5 |
> |87.5\% | -0.9 | -0.3 |
> ||
>
> **Q4:Why is the focus on “particularly the unconditioned or image-conditioned models?**
>
> Thanks. This sentence highlights our focus is not on the type of text-conditioned video generation methods that requires video description as input. This is because, understanding videos requires the model to analyze video content, whereas text-conditioned generation relies on descriptions that predefine the content. Such conflict makes it harder to unify these two approaches.
>
> **Q5:Minor weakness about the expressions.**
>
> Thank you so much for your carefully reading. We will definitely revise the expressions in the revision version.

---

> > ### Comment · Reviewer_PRtT · 2024-08-10
> >
> > I have read through the other reviews, and all of the authors' responses, and believe that the majority of the concerns have been addressed. In particular, the inclusion of sota methods, parameter counts, and timing for training and test, significantly improves Table 1. Again, this paper remains very interesting to me even if it doesn't beat sota, in particular because it shows "mutual enhancement" on the two tasks, which is a much-sought-after result.
> >
> > Thank you for double-checking the FVD computation. Your explanation makes sense. However, reviewer JBof suggests "for the generation ability evaluation, since the model is initialized from the SVD, it’s better to compare with SVD on different datasets". Should different baselines which are trained on significant video data be used for comparison instead?

---

> > > ### Author Response · Authors · 2024-08-11
> > >
> > > Thank you for thoroughly reviewing our work and responses. We appreciate your acknowledgment of the improvements, particularly with the inclusion of SOTA methods, parameter counts, and timing details. We’re glad you find our work interesting, and we share your enthusiasm about its potential implications.
> > >
> > > Regarding the SVD comparison, we agree that comparing with SVD directly provides a meaningful baseline. Thus, we have newly added such comparison. We conducted frame prediction tests using the SVD model on the SSV2 and Epic-Kitchen datasets. Since the SVD model performs better at higher resolutions, we generated videos at 512x512 resolution and then downsampled them to 256x256 for FVD calculations.
> > >
> > > The results show that while SVD achieves competitive scores compared to previous state-of-the-art methods, it is suboptimal compared to GenRec. This is likely due to SVD's design for general scenarios. Baseline I in Table 1 represents our enhanced SVD baseline, which fine-tunes SVD on the target datasets for better results and more a fairer comparison.
> > >
> > > Thank you again for your recognition and thoughtful feedback. We greatly appreciate your time and effort in reviewing our work. We will include the SVD baseline results in our revised version.
> > >
> > > |Method| SSv2 FVD | Epic FVD |
> > > |:--:|:--:|:--:|
> > > | **SVD** | **99.7** | **180.8** |
> > > | Baseline I | 50.3 | 53.6 |
> > > | GenRec | 46.5 | 49.3 |
> > > ||

---

### Official Review · Reviewer_caZh · 2024-07-14

**Soundness:** 2
**Presentation:** 2
**Contribution:** 2
**Rating:** 5
**Confidence:** 4

**Summary:**

This work proposes a video diffusion model to unifies the video generation and video recognition tasks. The proposed GenRec, during training, jointly optimize the generation objective and the classification objective. During inference, GenRec can deal with both the video generation conditioned on frames or classes, and video classification task. The experiments demonstrate that GenRec shows superior performance in both tasks.

**Strengths:**

- The proposed GenRec unifies the video generation and video recognition tasks.
- The experimental results validate the effectiveness of the proposed unified GenRec.

**Weaknesses:**

- Considering that previous works [8,37,46] have already demonstrated that diffusion models have the capability of extracting sufficient features for understanding tasks, the combination of generation and recognition tasks to boost both tasks is too straight-forward, limiting the novelty of the paper.
- It is unclear for the inference stage for video recognition. Like why the noised $\tilde{z}$ is still incorporated for classification, which may lead to the recognition result undeterministic. During inference, is the multi-step denoising procedure sill required?
- There is the lack of training/inference time cost comparison with previous video classification methods.

**Questions:**

Pleaser refer to the weakness part. I would like to modify my rating according to the responses from the authors.

**Limitations:**

The authors have already claimed the limitations.

---

> ### Author Rebuttal · Authors · 2024-08-06
>
> Thank you for your valuable comments. We will address your concerns point by point.
>
> **Q1: Concern about the novelty compared with [8,37,46].**
>
> Thanks. Previous works [8, 37, 46] applied diffusion models to understanding tasks, but there are some limitation that mainly differs from our paper:
> - [8] abandons generation capabilities in the end for improved discriminative performance;
> - [37] uses a frozen image diffusion model and only focuses specifically on low-level geometric correspondence;
> - [46] uses a frozen image diffusion backbone only for mask location prediction;
>
> In summary, they ignore visual generation, do not unify generation and recognition tasks, and also they are limited to the image domain. In contrast, our paper focuses on the video domain, addressing the following challenges:
>
> - We demonstrate that video diffusion models can leverage generative priors for analyzing high-level actions, especially for tasks requiring thorough temporal information analysis.
> - We demonstrate that video generation and understanding can coexist and enhance each other, improving recognition robustness with limited visual frames, and adherence to instructions for generation.
>
> We believe these explorations are novel and significant as suggested by Reviewer 96MS.
>
> [8] Deconstructing denoising diffusion models for self-supervised learning. arXiv preprint arXiv:2401.14404, 2024
>
> [37] Emergent correspondence from image diffusion. In Neurips, 2023
>
> [46] Open-vocabulary panoptic segmentation with text-to-image diffusion models. In CVPR, 2023
>
> **Q2: Why is noise incorporated during inference for video recognition, how does it affect result determinism, is multi-step denoising required during inference?**
>
> Thank you for your feedback. Our unified models are trained on noisy inputs, so maintaining consistency during inference is necessary to ensure robustness. Works [37, 46], as you previous mentioned, also need to add a certain-level noise before extracting visual features. We further do the testing on SSv2 action recognition for multiple times with different random seeds, shown as the table below. The results show that the accuracy is consistent with only 0.0125\% standard variation, indicating the model’s robustness to the noise. For fully deterministic results, the random seed for noise sampling should be fixed.
>
> During inference, only one single-step denoising is required for prediction. Thus, the inference process is efficient.
>
> |**Seed** | 0 | 1 | 2 | 3 | 4 | 5 | AVG|
> |:--:|:--:|:--:|:--:|:--:|:--:|:--:|:--:|
> |**SSv2 Acc** | 75.83\% | 75.82\% | 75.83\%  |75.83\%  |75.86\%  |75.84\%  |(75.835 $\pm$ 0.0125)\%  |
> | |
>
> **Q3: Training and inference time cost compared with previous classification methods.**
>
> Thank you for your valuable feedback. We compare the training and inference time cost between MVD-H, Baseline II and GenRec, fairly on the same hardware resource: 4-nodes * 8 V100s, and same batch size. Baseline II refers to fully fine-tuning the original SVD model with classification supervision only.
>
> As shown in the table, GenRec and the Baseline II consumes more time for testing than MVD-H. The difference in time primarily arises from the varying number of parameters. Our model is derived from a generative model, and the complexity of the generative task necessitates a larger number of parameters for effective learning. Compared with Baseline II, since GenRec requires additional decoder blocks for video generation training, the training time will get increased a little bit. As the additional up-sampling blocks will not be used when doing action recognition, GenRec shares the same testing time with Baseline II. It is worth noting that ''✝'' in MVD-H is to highlight that MVD method would use repeated augmentation technique during training, but such augmentation will significantly increase the training time. Our approach does not need to use that augmentation.
> All the methods do the down-stream finetuning for 30 epochs.
>
> |**Method** | Trainable Params | Total Params | Train Time | Train epochs | Test Time (2$\times$3 clips) |
> |:--:|:--:|:--:|:--:|:--:|:--:|
> |MVD-H ✝ |  633M | 633M  | 3038 s/Epoch| 30 | 441s |
> |Baseline II | 1.3B | 1.9B | 2954 s/Epoch | 30 | 1500s |
> | GenRec | 1.5B | 2.1B | 3751 s/Epoch | 30 | 1500s |
> | |

---

> > ### Author Response · Authors · 2024-08-14
> >
> > Dear Reviewer caZh,
> >
> > Hope this message finds you well. We wanted to kindly follow up on the feedback for our submission. Your insights are invaluable, and we would greatly appreciate any further comments you may have.
> >
> > Thank you for your time and consideration.

---

> ### Comment · Reviewer_caZh · 2024-08-14
>
> I have carefully read the responses from the authors and other reviewers' comments. The response has addressed my concerns, I would like to raise my rating to 5.

---

### Author Rebuttal · Authors · 2024-08-06

We thank the reviewers for their detailed, thoughtful and valuable feedback. We are grateful that the reviewers identified our work unifies the video generation and video recognition (Reviewer caZh), provides "exciting result" (Reviewer PRtT), are interesting and necessary for the computer vision (Reviewer JBof), are novel and helpful (Reviewer 96MS), are comprehensive and demonstrates strong performance (Reviewer C3Pd). We have tried our best to address all the questions and improve our paper following the guidance of the reviewers. If you have further questions, feel free to attach us by the official comment. The uploaded pdf contains a large table and some visualization cases, which are helpful for the following rebuttal responses.

---

### Comment · Area_Chair_R1UJ · 2024-08-13

Dear Reviewers,

We are nearing the end of the period for discussion with the authors. I kindly request you to review the comments made by other reviewers, as well as the responses from the authors.

Given that we have a mixed and divergent score (9, 7, 4, 4, 3), it's crucial that we have a comprehensive understanding of each other's viewpoints to make a fair and informed decision on the paper.

In particular, I would like to request Reviewers caZh, JBof, and C3Pd to actively participate in the discussion with the authors. Your insights and perspectives are highly valued and can significantly contribute to the decision-making process.

Thank you for your timely attention to this matter.

Best regards,
Area Chair

---

### Decision · Program_Chairs · 2024-09-25

**Decision:**

Accept (poster)

**Comment:**

This paper garnered overall positive reviews, with consensus on the merit of the proposed model that integrates video generation and recognition. The authors adeptly addressed concerns about novelty, experimental design, and efficiency. Given its innovative nature, compelling results, and satisfactory responses to feedback, I recommend the paper for publication.  It's a significant contribution that paves the way for future unified video generation and recognition research.